# Recent Advances in Genetics and Genomics of Snub-Nosed Monkeys (*Rhinopithecus*) and Their Implications for Phylogeny, Conservation, and Adaptation

**DOI:** 10.3390/genes14050985

**Published:** 2023-04-27

**Authors:** Weimin Kuang, Dietmar Zinner, Yuan Li, Xueqin Yao, Christian Roos, Li Yu

**Affiliations:** 1State Key Laboratory for Conservation and Utilization of Bio-Resource in Yunnan, School of Life Sciences, Yunnan University, Kunming 650500, Chinallliiiyuan@163.com (Y.L.); yaoxq1026@163.com (X.Y.); 2Cognitive Ethology Laboratory, German Primate Center, Leibniz Institute for Primate Research, 37077 Göttingen, Germany; dzinner@gwdg.de; 3Department of Primate Cognition, Georg-August-University of Göttingen, 37077 Göttingen, Germany; 4Leibniz-Science Campus Primate Cognition, 37077 Göttingen, Germany; 5Gene Bank of Primates, German Primate Center, Leibniz Institute for Primate Research, 37077 Göttingen, Germany; 6Primate Genetics Laboratory, German Primate Center, Leibniz Institute for Primate Research, 37077 Göttingen, Germany

**Keywords:** Colobinae, primates, phylogeny, phylogeography, demographic history, conservation genetics, adaptation

## Abstract

The snub-nosed monkey genus *Rhinopithecus* (Colobinae) comprises five species (*Rhinopithecus roxellana*, *Rhinopithecus brelichi*, *Rhinopithecus bieti*, *Rhinopithecus strykeri*, and *Rhinopithecus avunculus*). They are range-restricted species occurring only in small areas in China, Vietnam, and Myanmar. All extant species are listed as endangered or critically endangered by the International Union for Conservation of Nature (IUCN) Red List, all with decreasing populations. With the development of molecular genetics and the improvement and cost reduction in whole-genome sequencing, knowledge about evolutionary processes has improved largely in recent years. Here, we review recent major advances in snub-nosed monkey genetics and genomics and their impact on our understanding of the phylogeny, phylogeography, population genetic structure, landscape genetics, demographic history, and molecular mechanisms of adaptation to folivory and high altitudes in this primate genus. We further discuss future directions in this research field, in particular how genomic information can contribute to the conservation of snub-nosed monkeys.

## 1. Introduction

The rapid development in the field of molecular genetics during the past decade, particularly the advances in genome sequencing technologies, has led to an accumulation of genomic data for many species, including primates [1,2]. These data have the potential to refine phylogenetic and population genetic analyses [3,4] and broaden our understanding of adaptation and speciation [5]. These data can also have an impact on conservation planning and the preservation of endangered species [6,7,8].

Among primates, members of the genus *Rhinopithecus* (snub-nosed monkeys) are among the genetically most intensively studied taxa. To date, reference genomes are available for three species (ASM756505v1: *Rhinopithecus roxellana* [9]; ASM169854v2: *Rhinopithecus bieti* [10]; ASM2376470v1: *Rhinopithecus strykeri* [11]). In addition, more than one hundred re-sequenced genomes of *Rhinopithecus* have been published [10,12,13,14,15]. Snub-nosed monkeys are catarrhine primates of the family Cercopithecidae and the subfamily Colobinae. Together with langurs (*Presbytis*, *Trachypithecus*, and *Semnopithecus*) and the other odd-nosed monkey genera (*Pygathrix*, *Simias*, and *Nasalis*), they form the Asian colobine tribe Presbytini [16]. Snub-nosed monkeys are endemic to Southeast Asia. They inhabit subtropical and temperate mountain forests of southern China, northern Vietnam, and northern Myanmar [17,18].

The genus comprises five species (*Rhinopithecus roxellana* [*R. roxellana*], *Rhinopithecus bieti* [*R. bieti*], *Rhinopithecus brelichi* [*R. brelichi*], *Rhinopithecus strykeri* [*R. strykeri*], and *Rhinopithecus avunculus* [*R. avunculus*]) that diverged in the early Pleistocene [16,19,20,21,22]. *R. bieti* and *R. strykeri* are regarded as the Himalayan species, *R. avunculus* as the southern species, and *R. roxellana* and *R. brelichi* as the northern species. Fossils and molecular studies suggest that the genus most likely originated in the Hengduan Mountains in the border region of China and Myanmar [23,24,25]. While the ancestor of the Himalayan species remained in that region, the ancestor of the southern species dispersed to the southeast into Vietnam, and the ancestor of the northern species spread to central, northern, and southern China (Figure 1).

Fossils indicate that snub-nosed monkeys were widely distributed across Asia during the Pleistocene [23,26,27,28,29]. However, environmental changes during the Holocene led to habitat loss and fragmentation, which in turn led to a reduction in population size, particularly accelerated by the expansion of the human population within the last 400 years [30,31]. The wild population of *R. roxellana*, numbering ~22,500 individuals, is fragmented into three local populations in the Minshan and Qionglai Mountains (Sichuan/Gansu provinces; SG), the Qinling Mountains (Shanxi province; QL), and the Shennongjia National Nature Reserve (Hubei province; SNJ) [32] (Figure 1). Population sizes of the other species are much smaller (*R. bieti* ~3000 individuals [33] and *R. strykeri* ~950 individuals [34]). The species with the smallest population size and the highest risk of extinction are *R. avunculus* with less than 200 individuals [31] and *R. brelichi* with ~400 individuals [35]. All five species are among the world’s rarest and most endangered primates and are listed as Endangered or Critically Endangered by the International Union for Conservation of Nature (IUCN) [32].

Snub-nosed monkeys, like other colobine species, have a complex multi-chambered stomach similar to that of ruminants as an adaptation to their mainly folivorous diets [36,37]. They feed primarily on leaves and other non-fruit plant parts. Snub-nosed monkeys live at relatively high altitudes. *R. brelichi* and *R. avunculus* are found at elevations below 2000 m, and *R. roxellana*, *R. bieti,* and *R. strykeri* at higher elevations, i.e., above 3000 m. *Rhinopithecus bieti* and the gelada (*Theropithecus gelada*) are the only non-human primates that live in altitudes up to 4500 m [38,39,40,41,42].

Here, we provide an update on the phylogeny, phylogeography, population structure, landscape genetics, demographic history, and adaptation of snub-nosed monkeys based on recent genetic and genomic data. We further discuss future directions in this research field, in particular, how genomic information can contribute to the conservation of snub-nosed monkeys.

## 2. Phylogeny and Phylogeography

Genetic and genomic data revealed a basal phylogenetic position of *Rhinopithecus* within the odd-nosed monkeys, and they constitute the sister group to the combined *Pygathrix* and *Nasalis* + *Simias* clade [16,24]. Both fossil and molecular genetic evidence indicate that speciation occurred within only 1–2 million years [10,12,16,43,44]. As in many other phylogenetic reconstructions (e.g., *Papio*, *Chlorocebus*) [4,45,46,47,48], incongruences between phylogenies based on mitochondrial (mtDNA) and nuclear markers have been detected. After the discovery of *R. strykeri* in 2011 [22], Liedigk et al. [16] and Hong et al. [44] reconstructed phylogenies for all five species based on mitogenomes. They showed that the southern species (*R. avunculus*) diverged first and that the northern (*R. roxellana* and *R. brelichi*) species and the Himalayan (*R. bieti* and *R. strykeri*) species formed reciprocal monophyletic clades (Figure 2). However, whole-genome data, in agreement with a previous phylogeny based on several nuclear genes [16], strongly support the southern species as a sister taxon to the Himalayan clade, which together represent the sister clade to the two northern species. Possible explanations for these discordances could be either incomplete lineage sorting (ILS) and/or hybridization. Zhou et al. [12] estimated the amount of ILS affecting the genomes of snub-nosed monkeys at ~5.8%. However, gene flow became apparent as well between the ancestors of the Himalayan and northern species, and between the ancestor of the Himalayan species and *R. brelichi* (Figure 2) [12]. Thus, both ILS and historical (ancient) hybridization probably contributed to the phylogenetic incongruences observed in snub-nosed monkeys.

The estimated divergence times based on whole-genome data are more recent than the ones based on mtDNA data (Figure 2) [10]. Accordingly, the initial split, separating the northern species from the Himalayan and southern species, was dated at about 1.42 (1.05–1.81) million years ago (mya), and the divergence of the 2 northern species occurred about 1.07 (0.76–1.38) mya. The uplift of the Tibetan Plateau (Yuanmu movement ~1.60 mya) is most likely responsible for these speciation events [10]. The split between the southern species from the ancestor of the Himalayan species occurred at about 0.80 (0.58–0.99 mya), and the Himalayan species separated at about 0.44 (0.33–0.55) mya, coinciding with the Penultimate Glaciation (0.13–0.30 mya), indicating that this glacial period had an important impact on the divergence of the Himalayan species.

Besides these phylogenetic inconsistencies among snub-nosed monkey species, inconsistencies were also found in intraspecific analyses based on different genetic markers. For the three geographic populations of *R. roxellana* (SG, QL, and SNJ), matrilineal phylogenies based on mtDNA markers revealed a strong geographic pattern, reflecting the disjunct distribution of the populations [42,49,50]. In contrast, microsatellite-based analyses detected only two genetic clusters (SG/QL and SNJ) [51]. In a study combining mitogenomic, Y-chromosomal, and autosomal nuclear data, Kuang et al. [14] also obtained contradicting phylogenies. They found that repeated gene flow events among all three populations caused these incongruences.

For *R. bieti*, using a part of the mitochondrial control region (401 bp), Liu et al. [41,42] performed a population genetic analysis of eleven local groups and found evidence for a divide into two haplogroups (split time: 0.10–0.70 mya). In contrast, an analysis of ten microsatellite loci suggested a structuring into five subpopulations [41]. This finding was, however, not confirmed by analyses using single-nucleotide polymorphisms (SNPs) [10,13]. In the latter studies, no significant sub-structure was found, although the investigated samples probably did not cover the entire range of the species. Thus, large-scale population genome resequencing data are needed to further address the potential sub-structuring within *R. bieti*.

## 3. Population Genetics

Information on genetic diversity, genetic population structuring, degree of inbreeding, and demographic history is particularly important for endangered species close to extinction or species with declining population sizes [52,53]. Such information can help to develop conservation strategies [54]. Many population genetic studies using different marker systems have been performed on snub-nosed monkeys [10,13,14,15,40,41,42,49,50,51,55,56,57,58,59,60,61,62]. We will summarize the main findings below.

### 3.1. Genetic Diversity

Genetic diversity is one indicator of the adaptive potential of populations [63]. For instance, populations with higher genetic diversity have a greater potential for adapting to changing environments [64]. Compared to other endangered species, snub-nosed monkeys show generally low levels of genetic variation [40,49,50,57,58,59,60,61]. In the small population of *R. avunculus*, variability in mtDNA is extremely low and, in fact, the lowest mitochondrial genetic variability reported for any primate species in the wild to date [60]. Kuang et al. [15] combined all published genome data [10,12,13] for an investigation of 106 genomes representing all five species and assessed their genetic diversity. One measure of genetic diversity is the degree of heterozygosity (*He* = the number of heterozygous sites divided by the total number of callable sites across the whole genome). Within the five species, *He* ranged from 0.034 to 0.069%. Compared to other primates [65], snub-nosed monkey species exhibit extremely low levels of *He* (Figure 3). This finding is consistent with their small extant population sizes, which may be the result of historical bottlenecks due to severe fragmentation of and reduction in their habitats and populations [13]. Among snub-nosed monkey species, *R. brelichi* has the highest heterozygosity (*He* = 0.069%), followed by *R. roxellana* (*He* = 0.043%) and *R. avunculus* (*He* = 0.042%), while *R. bieti* (*He* = 0.034%) and *R. strykeri* (*He* = 0.036%) have the lowest values. In *R. roxellana*, the smallest local population (SNJ) has a relatively high degree of heterozygosity (*He* = 0.044%; Figure 3), similar to that of the largest local population (SG: *He* = 0.046%). The lowest degree of heterozygosity (*He* = 0.038%) was found in the QL population.

### 3.2. Landscape Genetics

Habitat fragmentation can lead to changes in population connectivity and thus into a reduction in gene flow [66]. This, in turn, can lead to a reduction in genetic diversity within populations and an elevation in genetic differentiation among populations [67,68,69]. Landscape genetics provides a set of tools to correlate the spatial heterogeneity of landscapes with estimations of gene flow, which helps to identify landscape features that significantly influence gene flow and can provide scientific support for the design of ecological (dispersal) corridors, and thus to develop conservation strategies [70,71].

All extant snub-nosed monkey species have small distribution ranges. Climatic changes and, more recently, human activities have led to habitat fragmentation and reduction, and also to limited inter-population exchange [30,31,72]. To date, in snub-nosed monkeys, landscape genetic analyses have been performed for *R. bieti* and *R. roxellana*. Liu et al. [41] obtained genetic samples from 135 individuals from 11 local groups across the geographical range of *R. bieti* and assessed their genetic structure using 10 microsatellite markers. Only 4.9% of the genetic distance was explained by geographic distance, while 36.2% of the genetic distance was explained by habitat gaps, mostly resulting from human activities, such as agricultural land use, highways, and settlements. These results suggest that the genetic diversity of *R. bieti* is negatively affected by habitat fragmentation due to anthropogenic landscape features. Li et al. [73] applied more models of landscape genetics to further investigate the effects of landscape configuration on gene flow among the local groups of *R. bieti*. They identified potential migration corridors among isolated local groups of *R. bieti*, with potentially higher connectivity in the northern part of the species range. Forest restoration in seven areas (between the central and southern parts of the range and between the southern groups) most likely will improve connectivity among local groups and thus the protection of *R. bieti*. Zhao et al. [33], using an integrated ecological and genetic approach, predicted the potential impact of climate change and future human activities on the species distribution and genetic patterns of *R. bieti* for the period 2000–2050. Human activities and climate change will have a strong negative impact on small, isolated local groups with low habitat connectivity. Human disturbances such as logging and livestock grazing need to be controlled in the region to effectively protect the species. For the three populations of *R. roxellana*, Zhao et al. [74] integrated species distribution models and genotype–environment association analyses to investigate the impact of climate change and human activities on these populations. Species distribution modelling showed that the area of suitable habit for *R. roxellana* decreased by 59.5% between the last interglacial (LIG) and the last glacial maximum (LGM), but expanded by 55.0% between the LGM and the present, indicating that this species experienced range expansion during warmer interglacial periods and range contraction during colder glacial periods. Using a genotype–environment association analysis, Zhao et al. [74] suggested that local adaptations to temperature and/or rainfall appear to have driven niche differentiation and genetic differences in the isolated subpopulations. The combined analysis of population genetics and landscape features can identify or confirm assumed causes for changes in the genetic diversity of the snub-nosed monkeys and the extent of gene flow.

### 3.3. Inbreeding

Inbreeding may result in inbreeding depression, which can cause a reduction in the adaptive potential of a population, ultimately leading to an increased risk of extinction [64,75,76]. Kuang et al. [15] estimated inbreeding levels in the snub-nosed monkeys and showed that the inbreeding coefficients (F_ROH_) of the 5 species ranged from 49.4 to 78.9%. *R. brelichi* has the lowest inbreeding coefficient (F_ROH_ = 49.4%), followed by *R. roxellana* (F_ROH_ = 68.1%) and *R. avunculus* (F_ROH_ = 68.2%). *R. strykeri* (F_ROH_ = 73.4%) and *R. bieti* (F_ROH_ = 78.9%) had higher inbreeding coefficients. Kuang et al. [15] assessed inbreeding levels by counting segments of runs of homozygosity (ROHs) greater than 1 Mb in length (ROH_1Mb_) in the nuclear genomes, which indicates recent inbreeding. Surprisingly, although the population size of *R. roxellana* was larger than that of the other four snub-nosed monkey species, they found that the ROH_1Mb_ of *R. roxellana* was significantly higher than that of the other four species. This suggests that inbreeding is more prevalent in *R. roxellana* compared to the other snub-nosed monkey species. The dramatic reduction in their distribution ranges over the last 400 years combined with a decline in population size might have caused the relatively high degree of inbreeding in snub-nosed monkeys [30].

### 3.4. Genetic Load

Genetic load is the actual or potential reduction in a population’s mean fitness due to genetic causes [77,78]. Species with small population sizes can possess deleterious alleles in high frequency, which can result in lowered individual fitness or even species extinction [79,80]. Zhou et al. [13] examined the genetic load in snub-nosed monkeys and found a similar number of derived missense and loss-of-function (LOF) alleles independent of the corresponding population size when comparing the five species. However, based on a larger sample size, Kuang et al. [15] found that individuals of *R. roxellana* carried significantly fewer deleterious variations than the other snub-nosed monkey species. High levels of inbreeding (see above) and relatively low levels of genetic load in *R. roxellana* suggest that inbreeding does not necessarily lead to a genome-wide enrichment in deleterious variation. Only 4.9% of all deleterious variations were detected in the long ROH regions, while the majority fell in the short ROH regions. This finding indicates that recent inbreeding in *R. roxellana* did not lead to an excessive accumulation of homozygous-derived deleterious variations, a finding that was recently also reported in snow leopards (*Panthera unica*) and island foxes (*Urocyon littoralis*) [81], and in mountain gorillas (*Gorilla beringei beringei*) [82]. In addition, the identified genes containing homozygous LOF variations are associated with immune-related functions in the genomes of snub-nosed monkeys. For instance, *F2RL3*, with an LOF variation in all five species, has been reported to be involved in immune cell responses and blood clotting behaviour [83,84]. Likewise, *CALML6*, *LCK*, and *MSR1* are related to immune regulation and play a key role in T- or B-cell responses and immune system homeostasis [85,86]. However, whether these deleterious variations affect immunity and viability in snub-nosed monkeys needs to be further verified by functional and physiological experiments.

### 3.5. Demographic History

Current patterns in population genetic variation have been shaped by both long-term evolutionary changes and contemporary demographic processes [87]. Reconstructing the past and recent demographic history of endangered species can be used to address questions concerning the causes of contemporary low genetic diversity, gene flow, and population differentiation extents, as well as population size fluctuations in species of conservation concern, which can be critical in guiding conservation efforts [88]. To identify recent and historic demographic events in snub-nosed monkeys, traditionally, mtDNA and microsatellites have been used [20,40,49,50,57,58,89,90]. For instance, based on the mtDNA control region, Luo et al. [50] and Liu et al. [40] inferred demographic histories of *R. roxellana* and *R. bieti*, respectively, revealing that they both experienced declines during the Last Glacial Maximum (LGM, the coldest multimillennial interval of the last glacial period). For *R. brelichi*, Kolleck et al. [58] found a clear decrease in effective population size (*Ne*) starting about 3500 to 4000 years ago, which coincides with the increasing human population in the area and the corresponding expansion of agriculture.

Although molecular approaches based on a single or a handful of markers have been successfully applied to model complex demographic histories and effective population sizes, genomic data provide greater statistical and analytical power. Using genome resequencing data, Zhou et al. [12,13] reconstructed the historical fluctuations in Ne for the Himalayan and northern species by a Pairwise Sequential Markovian Coalescent (PSMC) model. They found clear differences in the demographic history between *R. roxellana* and *R. brelichi* as well as the Himalayan species. According to their modelling, *R. roxellana* experienced two population expansions (~1.00 mya and ~0.05–0.07 mya) and two bottlenecks (~2 mya and ~0.10–0.40 mya), while in *R. brelichi*, *R. bieti*, and *R. strykeri*, a continuous decrease in *Ne* was observed. For the three populations of *R. roxellana*, Zhou et al. [13] and Kuang et al. [14] also inferred the recent demographic histories using genomic variance and demographic modelling. Their modelling revealed the division of the *R. roxellana* populations into two distinct genetic groups (SG/QL and SNJ similar to the microsatellite study of Luo et al. [51]). However, the divergence time of these two populations remains controversial. Kuang et al. [14] suggested that the split occurred at the beginning of the LGM, whereas Zhou et al. [13] fixed it near the end of the LGM. In addition, Kuang et al. [14] simulated the demographic development of the local populations of *R. roxellana* to investigate their origin and dispersal. Their modelling suggested that the ancestral *R. roxellana* population was once widespread and became fragmented but with continuous gene flow among the then-isolated subpopulations. This demographic scenario suggests that the ancestral bottleneck occurred during the Penultimate Glaciation (from ~194 kya to ~135 kya). The split into SNJ and non-SNJ (SG and QL populations), the population size reduction in the non-SNJ population, and the split between SG and QL within the non-SNJ population occurred around the beginning of the LGM. An extremely cold climate might have been the main cause of the severe population decline. The warmer Holocene might then have promoted the dispersal and expansion of the QL population, and the very recent decline in the SNJ population might be the result of human activities. In summary, climate changes during glacial and interglacial periods in the Pleistocene and Holocene (e.g., the Penultimate Glaciation and the LGM), as well as anthropogenic activities, have shaped the population history of *R. roxellana*.

## 4. Natural Selection and Adaptation

### 4.1. Adaptation to Folivory

Studies on dietary adaptation in non-human primates are important not only to understand how primates fill ecological niches, but also for comparative studies in human evolution and biomedical research [91]. Colobines are, due to their folivory, important model organisms for the study of primate dietary evolution [92,93]. In addition to anatomical and physiological adaptations to folivory, a growing number of studies have found that both the primate host and their gut microbes play a symbiotic role in dietary adaptation [94,95,96].

Previous studies focused on the genetic analysis of a digestive enzyme gene, the pancreatic ribonuclease (*RNASE1*), which is secreted from the pancreas and transported into the small intestine to degrade RNA [10,12,36,93,97,98]. Using both molecular analyses and functional assays, Yu et al. [10] found that similar to other colobine species, a duplication of the *RNASE1* gene occurred in snub-nosed monkeys, and its duplicated gene (*RNASE1B*) has evolved rapidly under positive selection to enhance ribonucleolytic activity in an altered microenvironment and increased the rate of bacterial RNA digestion. Based on analyses of gene families, Zhou et al. [12] found a substantial expansion in gene families that might be related to adaptations consistent with leaf consumption. These genes are significantly enriched for pathways related to xenobiotic biodegradation (e.g., thiamine metabolism, lysosome, and drug metabolism) and in salivary secretions, which most likely enhance the ability of snub-nosed monkeys to detoxify secondary compounds present in leaves. In addition, positively selected genes were also identified in snub-nosed monkeys, which are mainly associated with the elongation and biosynthesis of fatty acids, including propionate metabolism, pyruvate metabolism, and lipid binding, which are consistent with short-chain volatile acids (products of degradation of cellulose and hemicellulose) as the main energy source for snub-nosed monkeys. Interestingly, they also found that 61.7% of the olfactory genes in snub-nosed monkeys are pseudogenes, which may be related to their atrophied external nostrils and the degeneration of their olfactory system.

In addition to host-level genomic adaptations to folivory, the gut microbiome also plays a crucial role in snub-nosed monkeys. Previous studies have investigated the adaptation to folivory of snub-nosed monkeys at both the taxonomic and functional levels of the gut microbiome [11,99,100,101,102,103,104,105,106,107]. At the taxonomic level, *Bacillota* (utilize crude fibre) and *Bacteroidetes* (primary degraders of polysaccharides) were the most dominant bacteria in the gut microbiome of snub-nosed monkeys. Particularly, *Bacillota* with *Lachnospiraceae* and *Ruminococcaceae* families were the most abundant families in snub-nosed monkeys, a finding common to other studies on wild folivorous primates, possibly related to high-fibre consumption [100,101,102,103,104,105,106,107]. Microbes of the *Lachnospiraceae* and *Ruminococcaceae* families also contain enzyme transport mechanisms, and metabolic pathways that enable the degradation of complex plant material, such as cellulose, hemicellulose, and lignin. In addition to high *Bacillota* abundance, a low *Prevotella* to *Bacaeroides* (P/B) ratio also promotes the dietary fibre digestion of snub-nosed monkeys [101,103,105]. At the functional level, the functional annotation of metagenomics shows that the gut microbiota in snub-nosed monkeys is mainly involved in the metabolism and synthesis of lipids, carbohydrates, proteins, amino acids, glycan biosynthesis, and other secondary metabolites [103,106,107].

### 4.2. Adaptation to High Altitude

The genetic factors that allow organisms to cope with high altitude, i.e., low-oxygen and low-temperature environments, is an interesting topic in biological research. Among snub-nosed monkeys, both the Himalayan species and *R. roxellana* live at high altitudes, either on the edge of the Tibetan Plateau or in central China. Hence, snub-nosed monkeys provide an interesting primate model to study adaptation to high-altitude environments.

Yu et al. [108] analysed the mitogenomes of *R. roxellana* and *R. bieti* and found significant evidence for positive selection (amino acid residues in genes *NADH2* and *NADH6*) to high-altitude adaptation in *R. roxellana*, but not in *R. bieti*. However, genomic data revealed a more fine-grained picture of high-altitude adaptation. Using comparative genomics, Yu et al. [10] revealed that in *R. bieti*, genes in significantly expanded gene families were enriched in pathways related to DNA repair and damage response as well as oxidative phosphorylation processes, suggesting their adaption to increased exposure to ultraviolet radiation (UV) and energy metabolism requirements. In addition, RNA sequencing and comparative transcriptome analyses of multiple tissues from *R. bieti* showed that their highly expressed genes were enriched in pathways associated with oxidative phosphorylation and cardiac muscle contraction. Additional genomic analyses identified eight shared amino acid substitutions in six genes (*RNASE4*, *DNAH11*, *CDT1*, *RTEL1*, *ARMC2*, and *NT5DC1*) which are associated with lung function, DNA repair, and angiogenesis in the three species living at high altitudes (*R. bieti*, *R. strykeri*, and *R. roxellana*). UV irradiation experiments on *CDT1* (Ala537Val), a gene associated with DNA repair, showed that the phenotype with the mutated variant is more stable compared to the wild type, suggesting that the mutated version of the gene contributes to the resistance against UV light in high-altitude environments. In addition, mutations (Asn89Lys and Thr128Ile) in the angiogenesis-related *RNASE4* gene may lead to increased activity in inducing tubular structures in human umbilical vein endothelial cells (HUVEC) and could enhance the angiogenic ability of *RNASE4* and thus help snub-nosed monkeys to adapt to high-altitude environments.

In addition, Yu et al. [10] detected candidate genes under selection in *R. bieti* and *R. roxellana* populations. These genes were enriched in signalling pathways and cellular functions that were particularly associated with DNA repair, cardiac and vascular development, hypoxic response, energy metabolism, and angiogenesis. Positively selected genes were also significantly enriched in biological processes related to chromosome structural stability and DNA metabolism in both *R. bieti* and *R. roxellana*. For example, *BRCC3* and *RYR2* play important regulatory roles in DNA damage repair and cardiac function. Of these two genes, *RYR2* was also found to be positively selected in another high-altitude species, the Tibetan grey wolf (*Canis lupus chanco*) [109]. Other positively selected genes were associated with vascular function and development (*NOX1* and *HDAC9*), DNA damage response repair (*SETD2*), cardiac function development (*CASQ2*), and energy metabolism (*PDK3* and *ACSS3*). In addition, Zhou et al. [13] identified several genes associated with high-altitude adaptation in *R. bieti*. Of these positively selected genes, *ADAM9*, a member of the *ADAM* family, encodes for a cell-surface membrane glycoprotein, which has a regulatory function in the cellular hypoxic response. This gene family is also under positive selection in high-altitude species (e.g., *Bos grunniens* [110] and *Pseudopodoces humilis* [111]). In addition, another two candidate genes, *HERC1* and *SLC9A6*, are in the pathway of cellular hypoxic response and may be associated with high-altitude acclimation in *R. bieti*. Overall, Yu et al. [10] and Zhou et al. [13] detected diverse genes under selection that might be related to high-altitude adaptation, specifically to hypothermia, low oxygen, and intensive UV light exposure.

## 5. Conclusions and Prospects

Here, we reviewed the recent major advances in the genetics and genomics of snub-nosed monkeys concerning their phylogeny, population genetics, and adaptive evolution. These advances are also of significance for the conservation of snub-nosed monkeys.

With the development in genome sequencing technology, the study of conservation genetics has entered the era of conservation genomics. Genetic markers have developed from the original short mitochondrial segments and a few microsatellite loci to thousands of genome-wide SNPs [53,112]. In addition to abundant SNPs, other types of genomic markers such as insertions/deletions (INDELS, which are generally defined as <50 bp in length), structural variants (SVs, which are generally defined as >50 bp in length), and even variation in chromosomal structure [113] are also needed for systematic and comprehensive comparative studies. In addition, previous studies have investigated the adaptation processes (e.g., adaptation to folivory and high altitudes) and conservation genetics of snub-nosed monkeys based on protein-coding regions [10,12] and genomic non-coding regions, such as conserved non-coding elements (CNEs, which are involved in the regulation of gene expression) [114]. Such regions of the genome should be increasingly investigated to further elucidate the molecular mechanisms of the adaptive evolution and conservation of snub-nosed monkeys from a regulatory perspective.

For endangered primate species, such as snub-nosed monkeys, sample collection from wild populations is sometimes difficult, and often, non-invasively collected material such as faeces and hair is the only source for population genetic studies. However, endogenous DNA extracted from such material is generally of low quantity and quality, making whole-genome sequencing difficult. Therefore, there is an urgent need to overcome the technical bottleneck, aiming to obtain a large amount of good-quality DNA from non-invasively collected samples for population genomic analysis in future studies. Technical developments in the extraction and enrichment of ancient DNA from archaeological material or museum samples, such as hybridization capture and methyl-CpG-blinding domain (MBD) enrichment techniques, or fluorescence-activated cell sorting from faecal material (fecalFACS) [115,116,117,118], provide effective methods to increase the proportion of the targeted endogenous DNA. Future studies will apply such techniques, aiming to generate large amounts of population genomic data. This will further contribute to our understanding of the fine-scale population structures and comprehensive demographic histories of snub-nosed monkeys.

It would be also interesting to investigate genomically sub-fossil and historical (museum) specimens of snub-nosed monkeys that are decades, hundreds, or thousands of years old. Such studies will greatly expand our knowledge about changes in genetic diversity over time, evolutionary and demographic history, and how human activities influenced these processes in these enigmatic primate species.

## Figures and Tables

**Figure 1 genes-14-00985-f001:**
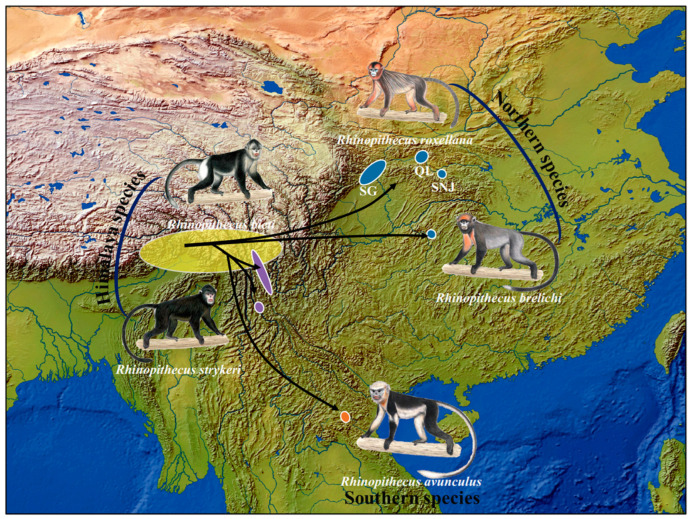
Geographic distribution of snub-nosed monkeys (*Rhinopithecus* spp.) and possible historical dispersal routes (black curves) from their likely geographic origin in the Hengduan Mountains (yellow ellipsis). Other ellipses denote the current distribution of southern (*R. avunculus*; orange), northern (*R. roxellana* and *R. brelichi*; blue), and Himalayan (*R. bieti* and *R. strykeri*; purple) species, respectively. Drawings by Stephan Nash, used with permission.

**Figure 2 genes-14-00985-f002:**
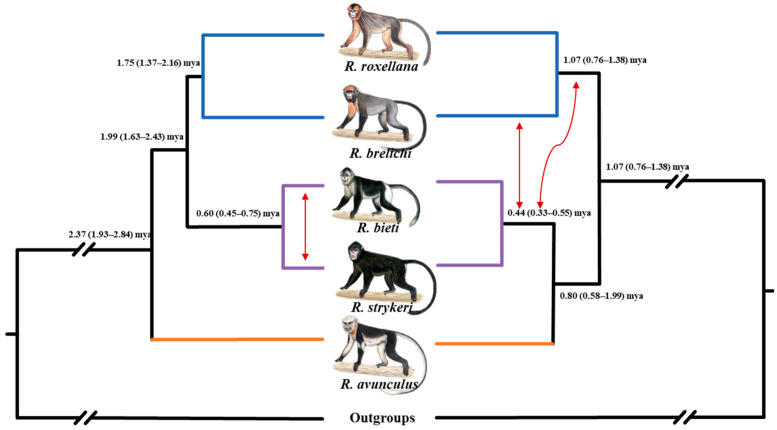
Phylogeny and divergence ages of *Rhinopithecus* based on mitogenome data (left, data from [16]) and whole-genome data (right, data from [12]). Orange, blue, and purple branches represent the southern, northern, and Himalayan species, respectively. The red arrows indicate historical gene flow events. The estimated divergence times are given in million years ago (mya) along with their 95% confidence intervals.

**Figure 3 genes-14-00985-f003:**
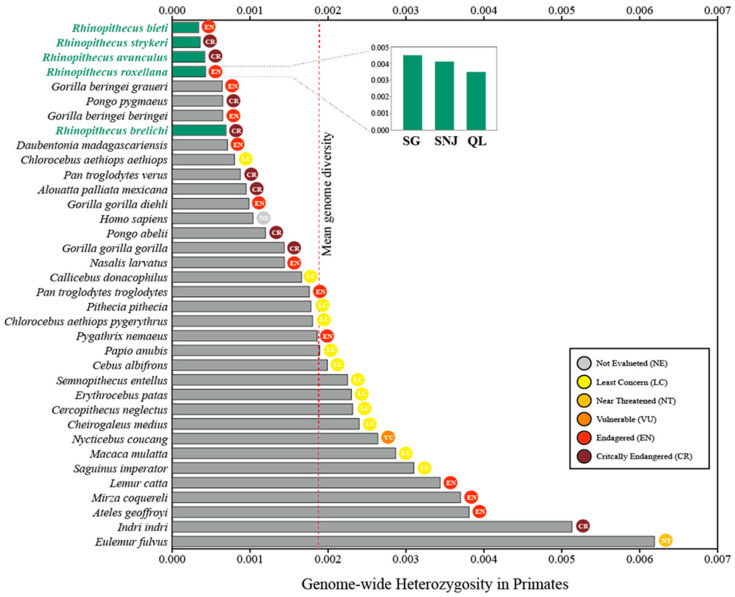
Genome-wide heterozygosity of *Rhinopithecus* species (data from [15]) compared to other primate species [65]. Also given are the categories of the IUCN Red List.

## Data Availability

All data are publicly available from the cited references.

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
