# Peer review of "Recent Advances in Genetics and Genomics of Snub-Nosed Monkeys (Rhinopithecus) and Their Implications for Phylogeny, Conservation, and Adaptation"

_genes, 2023, doi:10.3390/genes14050985_

Round 1

Reviewer 1 Report

Overall, the review of snub-nosed monkeys' genetics and genomics is well-written and presented. The outcomes of this study are given in a less detailed manner in the abstract, which should be updated. In addition, I propose incorporating an overall representation of genetic landscape shape interpolation analysis using existing genetic data and species distribution modeling, niche overlapping, and/or suitable habit (past, present, and future) by using existing occurrence records to discuss and reinforce the raised biological questions, particularly the landscape genetics, demographic history, and adaptation of these primate species.

Author Response

Dear Reviewer,

Thank you for your decision and construction comments on our manuscript. We have carried out a revision of our manuscript according to the comments. Please find my itemized responses in below and our revision/correction in the re-submitted files.

Best wishes,

Weimin Kuang

Comments: Overall, the review of snub-nosed monkeys' genetics and genomics is well-written and presented. The outcomes of this study are given in a less detailed manner in the abstract, which should be updated. In addition, I propose incorporating an overall representation of genetic landscape shape interpolation analysis using existing genetic data and species distribution modeling, niche overlapping, and/or suitable habit (past, present, and future) by using existing occurrence records to discuss and reinforce the raised biological questions, particularly the landscape genetics, demographic history, and adaptation of these primate species.

Response: Thank you for your positive comments and constructive suggestions. We changed the abstract slightly, but did not provide detailed results in the abstract since the presentation and discussion of the published results are the focus of the review itself. We believe that the last two sentences of our abstract “Here we review recent major advances in snub-nosed monkey genetics and genomics and their impact on our understanding of the phylogeny, phylogeography, population genetic structure, landscape genetics, demographic history, and molecular mechanisms of adaptation to folivory and high-altitude in this primate genus. We further discuss future directions in this research field, in particular how genomic information can contribute to the conservation of snub-nosed monkeys.” nicely summarize the topics and aims of our review.

We agree that integrating species distribution modeling and/or a genotype-environment analysis would be a nice addition. However, since our paper is a review article presenting genetic and genomic information of snub-nosed monkeys published to date, we refrained from conducting ecological niche or species distribution analysis. This would be well beyond the scope of our review. However, we updated our paper by integrating results of a recent publication on landscape genetics of the Sichuan snub-nosed monkey, which may address your concerns (see lines 209-218 in the revised manuscript).

Reviewer 2 Report

This is an excellent review of the genetics and genomics of snub-nosed monkeys, with substantial information about evolutionary and population genetics of these rare species.  The review is particularly detailed regarding the difficulties of resolving potential conflicts raised by different levels of genomic variation, and also provides a sophisticated discussion of alternative explanations of range and speciation.  

My only possible complaint regards the penultimate paragraph in the discussion which, in my opinion, is somewhat meandering.

Author Response

Dear Reviewer,

Thank you for your decision and construction comments on our manuscript. We have carried out a revision of our manuscript according to the comments. Please find my itemized responses in below and our revision/correction in the re-submitted files.

Best wishes,

Weimin Kuang

Comments: This is an excellent review of the genetics and genomics of snub-nosed monkeys, with substantial information about evolutionary and population genetics of these rare species.  The review is particularly detailed regarding the difficulties of resolving potential conflicts raised by different levels of genomic variation, and also provides a sophisticated discussion of alternative explanations of range and speciation.

My only possible complaint regards the penultimate paragraph in the discussion which, in my opinion, is somewhat meandering.

Response: Thank you for your positive comments and constructive suggestions. We revised the penultimate paragraph in the discussion (see lines 418-432 in the revised manuscript).

Reviewer 3 Report

Evaluation of the manuscript entitled ” Recent advances in genetics and genomics of snub-nosed monkeys (Rhinopithecus) and their implications for phylogeny, conservation, and adaptation   “ submitted by Kuang et al.

The manuscript is a review of the genetics and genomics of snub-nosed monkey species with particular focus on their phylogeny, population genetics, and adaptive evolution. The manuscript contains valuable information about genetic diversity, inbreeding in small populations, genetic load, and demographic history, which are of great significance for plans of the conservation of snub-nosed monkeys. The chapters about dietary adaptation to folivory and adaptation to high altitude are very interesting and provide many examples of positive selection in genes associated with physiological function. The manuscript is well organized and well written. In conclusion, I therefore recommend acceptance of the manuscript.

Author Response

Comments: Evaluation of the manuscript entitled “Recent advances in genetics and genomics of snub-nosed monkeys (Rhinopithecus) and their implications for phylogeny, conservation, and adaptation” submitted by Kuang et al.

The manuscript is a review of the genetics and genomics of snub-nosed monkey species with particular focus on their phylogeny, population genetics, and adaptive evolution. The manuscript contains valuable information about genetic diversity, inbreeding in small populations, genetic load, and demographic history, which are of great significance for plans of the conservation of snub-nosed monkeys. The chapters about dietary adaptation to folivory and adaptation to high altitude are very interesting and provide many examples of positive selection in genes associated with physiological function. The manuscript is well organized and well written. In conclusion, I therefore recommend acceptance of the manuscript.

Response: Thank you for your positive comment and for supporting our manuscript.

Reviewer 4 Report

Dear Editor,

The review on Rhinopithecus genetics and genomics is an extensive and comprehensive work. The content is great and touches on many aspects of phylogeography, population structure, and adaptations. It is a pleasure to read it. I do not have any particular comments. The review can be published as is.

Author Response

Comments: The review on Rhinopithecus genetics and genomics is an extensive and comprehensive work. The content is great and touches on many aspects of phylogeography, population structure, and adaptations. It is a pleasure to read it. I do not have any particular comments. The review can be published as is.

Response: Thank you for your positive comment and for supporting our manuscript. Your suggested changes were all incorporated.